# The Sample Complexity of Semi-Supervised Learning with Nonparametric Mixture Models

**Chen Dan**[1], **Liu Leqi**[1], **Bryon Aragam**[1], **Pradeep Ravikumar**[1], **Eric P. Xing**[1,2]
[1]Carnegie Mellon University    [2]Petuum Inc.
{cdan,leqil,naragam,pradeepr,epxing}@cs.cmu.edu

## Abstract

We study the sample complexity of semi-supervised learning (SSL) and introduce new assumptions based on the mismatch between a mixture model learned from unlabeled data and the true mixture model induced by the (unknown) class conditional distributions. Under these assumptions, we establish an $\Omega(K \log K)$ labeled sample complexity bound without imposing parametric assumptions, where $K$ is the number of classes. Our results suggest that even in nonparametric settings it is possible to learn a near-optimal classifier using only a few labeled samples. Unlike previous theoretical work which focuses on binary classification, we consider general multiclass classification ($K > 2$), which requires solving a difficult permutation learning problem. This permutation defines a classifier whose classification error is controlled by the Wasserstein distance between mixing measures, and we provide finite-sample results characterizing the behaviour of the excess risk of this classifier. Finally, we describe three algorithms for computing these estimators based on a connection to bipartite graph matching, and perform experiments to illustrate the superiority of the MLE over the majority vote estimator.

## 1 Introduction

With the rapid growth of modern datasets and increasingly passive collection of data, labeled data is becoming more and more expensive to obtain while unlabeled data remains cheap and plentiful in many applications. Leveraging unlabeled data to improve the predictions of a machine learning system is the problem of semi-supervised learning (SSL), which has been the source of many empirical successes [1–3] and theoretical inquiries [4–16]. Commonly studied assumptions include identifiability of the class conditional distributions [5, 6], the cluster assumption [10, 11] and the manifold assumption [9, 12, 13, 15]. In this work, we propose a new type of assumption that loosely combines ideas from both the identifiability and cluster assumption perspectives. Importantly, we consider the general multiclass ($K > 2$) scenario, which introduces significant complications. In this setting, we study the sample complexity and rates of convergence for SSL and propose simple algorithms to implement the proposed estimators.

The basic question behind SSL is to connect the marginal distribution over the unlabeled data $\mathbb{P}(X)$ to the regression function $\mathbb{P}(Y \mid X)$. We consider multiclass classification, so that $Y \in \mathcal{Y} = \{\alpha_1, \ldots, \alpha_K\}$ for some $K \geq 2$. In order to motivate our perspective, let $F^*$ denote the marginal density of the unlabeled samples and suppose that $F^*$ can be written as a mixture model

$$F^*(x) = \sum_{b=1}^{K} \lambda_b f_b(x). \tag{1}$$

Assuming the unlabeled data can be used to learn the mixture model (1), the question becomes *when is this mixture model useful for predicting Y ?* Figure 1 illustrates an idealized example.

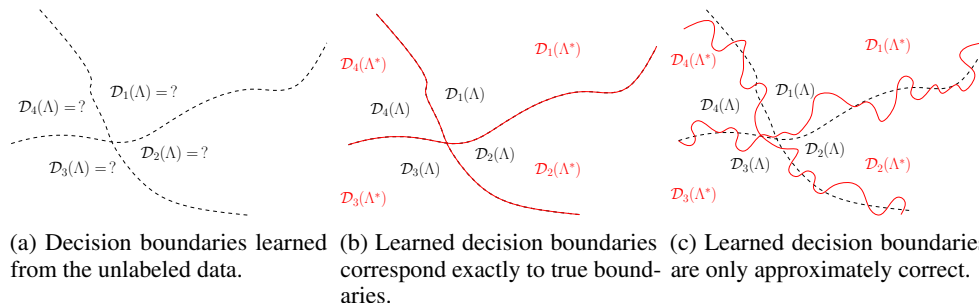

(a) Decision boundaries learned from the unlabeled data.

(b) Learned decision boundaries correspond exactly to true boundaries.

(c) Learned decision boundaries are only approximately correct.

Figure 1: Illustration of the main idea for $K = 4$. The decision boundaries learned from the unlabeled data (cf. (1)) are depicted by the dashed black lines and the true decision boundaries are depicted by the solid red lines. (a) The unlabeled data is used to learn some approximate decision boundaries via the mixture model $\Lambda$. Even with these decision boundaries, it is not known which class each region corresponds to. The labeled data is used to learn this assignment. (b) Previous work assumes that the true and learned decision boundaries are the same. (c) In the current work, we assume that the true decision boundaries are unknown, but that it is be possible to learn a mixture model that approximates the true boundaries using unlabeled data.

Clearly, we can always write $F^*(x) = \sum_{k=1}^{K} \lambda_k^* f_k^*(x)$, where $f_k^*$ is the density of the $k$th class conditional $\mathbb{P}(X \mid Y = \alpha_k)$ and $\lambda_k^* = \mathbb{P}(Y = \alpha_k)$. This will be called the *true mixture model* in the sequel. In this work, we *do not* assume that (1) is the true mixture model, i.e. we do not assume that $f_b$ corresponds to one of the $f_k^*$, nor do we assume that $\lambda_b$ corresponds to some $\lambda_k^*$. In other words, we allow the true mixture model to be nonidentifiable and consider the case where some misspecified mixture model is learned from the unlabeled data. We assume that the number of mixture components $K$ is the same as the number of classes, which is always known, although extending our analysis to overfitted mixtures is straightforward.

In an early series of papers, Castelli and Cover [5, 6] considered this question under the following assumptions: (a) For each $b$ there is some $k$ such that $f_b = f_k^*$ and $\lambda_b = \lambda_k^*$, (b) $F^*$ is known, and (c) $K = 2$. Thus, they assumed that the true components and weights were known but it was unknown which class each mixture component represents. In Figure 1, this corresponds to the case (b) where the true and learned decision boundaries are identical. Given labeled data, the special case $K = 2$ reduces to a simple hypothesis testing problem which can be tackled using the Neyman-Pearson lemma. In this paper, we are interested in settings where each of these three assumptions fail:

(a) *What if the class conditionals $f_k^*$ are unknown?* Although we can always write $F^*(x) = \sum_k \lambda_k^* f_k^*(x)$, it is generally not the case that this mixture model is learnable from unlabeled data alone. In practice, what is learned will be different from this ideal case, but the hope is that it will still be useful. In this case, the argument in Castelli and Cover [5] breaks down. Motivated by recent work on nonparametric mixture models [17], we study the general case where the true mixture model is not known or even learnable from unlabeled data.

(b) *What if $F^*$ is unknown?* In a follow-up paper, Castelli and Cover [6] studied the case where $F^*$ is unknown by assuming that $K = 2$ and the class conditional densities $\{f_1^*, f_2^*\}$ are known up to a permutation. In this setting, the unlabeled data is used to ascertain the relative mixing proportions, but estimation error in the densities is not considered. We are interested in the general case in which a finite amount of unlabeled data is used to estimate both the mixture weights and densities.

(c) *What if $K > 2$?* If $K > 2$, once again the argument in Castelli and Cover [5] no longer applies, and we are faced with a challenging permutation learning problem. Permutation learning problems have gained notoriety recently owing to their applicability to a wide variety of problems, including statistical matching and seriation [18–20], graphical models [21, 22], and regression [23, 24], so these results may be of independent interest.

With these goals in mind, we study the MLE and majority voting (MV) rules for learning the unknown class assignment introduced in the next section. Our assumptions for MV are closely related to recent work based on the so-called cluster assumption [4, 10, 11, 25]; see Section 4.2 for more details.

**Contributions** A key aspect of our analysis is to establish conditions that connect the mixture model (1) to the true mixture model. Under these conditions we prove nonasymptotic rates of convergence for learning the class assignment (Figure 1a) from labeled data when $K > 2$, establish an $\Omega(K \log K)$ sample complexity for learning this assignment, and prove that the resulting classifier converges to the Bayes classifier. We then propose simple algorithms based on a connection to bipartite graph matching, and illustrate their performance on real and simulated data.

## 2  SSL as permutation learning

In this section, we formalize the ideas from the introduction using the language of mixing measures. We adopt this language for several reasons: 1) It makes it easy to refer to the parameters in the mixture model (1) by wrapping everything into a single, coherent statistical parameter $\Lambda$, 2) We can talk about convergence of these parameters via the Wasserstein metric, and 3) It simplifies discussions of identifiability in mixture models. Before going into technical details, we summarize the main idea as follows (see also Figure 1):

1. Use the unlabeled data to learn a $K$-component mixture model that approximates $F^*$, which is represented by the mixing measure $\Lambda$ defined below;
2. Use the labeled data to determine the correct assignment $\pi$ of classes $\alpha_k$ to the decision regions $\mathcal{D}_b(\Lambda)$ defined by $\Lambda$;
3. Based on the pair $(\Lambda, \pi)$, define a classifier $g_{\Lambda, \pi} : \mathcal{X} \to \mathcal{Y}$ by (3) below.

**Mixing measures and mixture models** For concreteness, we will work on $\mathcal{X} = \mathbb{R}^d$, however, our results generalize naturally to any space $\mathcal{X}$ with a dominating measure and well-defined density functions. Let $\mathcal{P} = \{f \in L^1(\mathbb{R}^d) : \int f \, dx = 1\}$ be the set of probability density functions on $\mathbb{R}^d$, and $\mathcal{M}_K(\mathcal{P})$ denote the space of probability measures over $\mathcal{P}$ with precisely $K$ atoms. An element $\Lambda \in \mathcal{M}_K(\mathcal{P})$ is called a *(finite) mixing measure*, and can be thought of as a convenient mathematical device for encoding the weights $\{\lambda_k\}$ and the densities $\{f_k\}$ into a single statistical parameter. By integrating against this measure, we obtain a new probability density which is denoted by

$$m(\Lambda) := \sum_{b=1}^{K} \lambda_b f_b(x), \tag{2}$$

where $f_b$ is a particular enumeration of the densities in the support of $\Lambda$ and $\lambda_b$ is the probability of the $b$th density. Thus, (1) can be written as $F^* = m(\Lambda)$. By metrizing $\mathcal{P}$ via the total variation distance $d_{\text{TV}}(f, g) = \frac{1}{2} \int |f - g| \, dx$, the distance between two finite $K$-mixtures can be computed via the Wasserstein metric [26]:

$$W_1(\Lambda, \Lambda') = \inf \left\{ \sum_{i,j} \sigma_{ij} d_{\text{TV}}(f_i, f_j') : 0 \leq \sigma_{ij} \leq 1, \sum_{i,j} \sigma_{ij} = 1, \sum_{i} \sigma_{ij} = \lambda_j', \sum_{j} \sigma_{ij} = \lambda_i \right\}.$$

**Decision regions, assignments, and classifiers** Any mixing measure $\Lambda$ defines $K$ decision regions given by $\mathcal{D}_b = \mathcal{D}_b(\Lambda) := \{x \in \mathcal{X} : \lambda_b f_b(x) > \lambda_j f_j(x) \, \forall j \neq b\}$ (Figure 1). This allows us to assign an index from $1, \ldots, K$ to any $x \in \mathcal{X}$, and hence defines a function $\breve{g}_\Lambda : \mathcal{X} \to [K] := \{1, \ldots, K\}$. This function is not a genuine classifier, however, since its output is an uninformative index $b \in [K]$ as opposed to a proper class label $\alpha_k \in \mathcal{Y}$. The key point is that even if we know $\Lambda$, we still must identify each label $\alpha_k$ with a decision region $\mathcal{D}_b(\Lambda)$, i.e. we must learn a permutation $\pi : \mathcal{Y} \to [K]$. With some abuse of notation, we will sometimes write $\pi(k)$ instead of $\pi(\alpha_k)$ for any permutation $\pi$. Together, any pair $(\Lambda, \pi)$ defines a classifier $g_{\Lambda, \pi} : \mathcal{X} \to \mathcal{Y}$ by

$$g_{\Lambda, \pi}(x) = \pi(\breve{g}_\Lambda(x)) = \sum_{b=1}^{K} \pi^{-1}(b) \mathbf{1}(x \in \mathcal{D}_b(\Lambda)). \tag{3}$$

This mixing measure perspective helps to clarify the role of the unknown permutation in supervised learning: The unlabeled data is enough to learn $\Lambda$ (and hence the decision regions $\mathcal{D}_b(\Lambda)$), however, labeled data are necessary to learn an assignment $\pi$ between classes and decision regions. This formulates SSL as a coupled mixture modeling and permutation learning problem: Given unlabeled and labeled data, learn a pair $(\widehat{\Lambda}, \widehat{\pi})$ which yields a classifier $\widehat{g} = g_{\widehat{\Lambda}, \widehat{\pi}}$.

**Bayes classifiers and the true permutation**    The true permutation $\pi^* : \mathcal{Y} \to [K]$ is defined to be the permutation that assigns each class $\alpha_k$ to the correct decision region $\mathcal{D}_b^* = \mathcal{D}_b(\Lambda^*)$ (Figure 1). As usual, the target classifier is the *Bayes classifier*, which can also be written in the form (3): Let $\Lambda^*$ denote the true mixing measure that assigns probability $\lambda_k^*$ to the density $f_k^*$ and note that $F^* = m(\Lambda^*)$, which is the *true mixture model* defined previously. Then it is easy to check that $g_{\Lambda^*, \pi^*}$ is the Bayes classifier.

**Identifiability**    Although the true mixing measure $\Lambda^*$ may not be identifiable from $F^*$, some other mixture model may be. In other words, although it may not be possible to learn $\Lambda^*$ from unlabeled data, it may be possible to learn some other mixing measure $\Lambda \neq \Lambda^*$ such that $m(\Lambda) = F^* = m(\Lambda^*)$ (Figure 1c). This essentially amounts to a violation of the cluster assumption: High-density clusters are identifiable, but in practice the true class labels may not respect the cluster boundaries. Assumptions that guarantee a mixture model are identifiable are well-studied [27–29], including both parametric [30] and nonparametric [17, 31, 32] assumptions. In particular, Aragam et al. [17] have proved general conditions under which mixture models with arbitrary, overlapping nonparametric components are identifiable and estimable, including extreme cases where each component $f_k$ has the same mean. Since this problem is well-studied, we focus hereafter on the problem of learning the permutation $\pi^*$. Thus, in the sequel we will assume that we are given an arbitrary mixing measure $\Lambda$ which will be used to estimate $\pi^*$. We do not assume that $\Lambda = \Lambda^*$ or even that these mixing measures are close: The idea is to elicit conditions on $\Lambda$ that ensure consistent estimation of $\pi^*$. This makes our analysis applicable to a wide variety of methods, including heuristic approaches for learning $\Lambda$ from the unlabeled data, which are common in the literature on nonparametric mixtures.

## 3   Two estimators

Assume we are given a mixing measure $\Lambda$ along with the labeled samples $(X^{(i)}, Y^{(i)}) \in \mathcal{X} \times \mathcal{Y}$. Two natural estimators of $\pi^*$ are the MLE and majority vote. Although both estimators depend on $\Lambda$, this dependence will be suppressed for brevity.

**Maximum likelihood**    Define $\ell(\pi; \Lambda, X, Y) := \log \lambda_{\pi(Y)} f_{\pi(Y)}(X)$. We will work with the following *misspecified* MLE (i.e. $\Lambda \neq \Lambda^*$):

$$\widehat{\pi}_{\text{MLE}} \in \arg\max_{\pi} \ell_n(\pi; \Lambda), \quad \ell_n(\pi; \Lambda) := \frac{1}{n} \sum_{i=1}^{n} \ell(\pi; \Lambda, X^{(i)}, Y^{(i)}). \tag{4}$$

When $\Lambda = \Lambda^*$, this is the correctly specified MLE of the unknown permutation $\pi^*$, however, the definition above allows for the general misspecified case $\Lambda \neq \Lambda^*$.

**Majority vote**    The majority vote estimator (MV) is given by a simple majority vote over each decision region. Formally, we define a permutation $\widehat{\pi}_{\text{MV}}$ as follows: The inverse assignment $\widehat{\pi}_{\text{MV}}^{-1} : [K] \to \mathcal{Y}$ is defined by

$$\widehat{\pi}_{\text{MV}}^{-1}(b) = \arg\max_{\alpha \in \mathcal{Y}} \sum_{i=1}^{n} \mathbb{1}(Y^{(i)} = \alpha, X^{(i)} \in \mathcal{D}_b(\Lambda)). \tag{5}$$

If there is no majority class in a given decision region, we consider this a failure of MV and treat it as undefined. Note that when $K = 2$, the MV classifier defined by (3) with $\pi = \widehat{\pi}_{\text{MV}}$ is essentially the same as the three-step procedure described in Rigollet [10], which focuses on bounding the excess risk under the cluster assumption. In contrast, we are interested in the consistency of the unknown permutation $\pi^*$ when $K > 2$, which is a more difficult problem.

## 4   Statistical results

Our main results establish rates of convergence for both the MLE and MV introduced in the previous section. We will use the notation $\mathbb{E}_* h(X, Y)$ to denote the expectation with respect to the true distribution $(X, Y) \sim \mathbb{P}(X, Y)$. Without loss of generality, we assume that $\pi^*(\alpha_k) = k$ and $f_b = f_b^* + h_b$ for some $h_b$. Then $\widehat{\pi} = \pi^*$ if and only if $\widehat{\pi}(\alpha_k) = k$, which helps to simplify the notation in the sequel.

## 4.1 Maximum likelihood

Given $\Lambda$, the notation $\mathbb{E}_*\ell(\pi;\Lambda,X,Y) = \mathbb{E}_* \log \lambda_{\pi(Y)} f_{\pi(Y)}(X)$ denotes the expectation of the *misspecified* log-likelihood with respect to the *true* distribution. Define the "gap"

$$\Delta_{\mathrm{MLE}}(\Lambda) := \mathbb{E}_*\ell(\pi^*;\Lambda,X,Y) - \max_{\pi \neq \pi^*} \mathbb{E}_*\ell(\pi;\Lambda,X,Y). \tag{6}$$

For any function $a : \mathbb{R} \to \mathbb{R}$, define the usual Fenchel-Legendre dual $a^*(t) = \sup_{s \in \mathbb{R}}(st - a(s))$. Let $U_b = \log \lambda_b f_b(X)$ and $\beta_b(s) = \log \mathbb{E}_* \exp(sU_b)$. Finally, let $n_k := |\{i : Y^{(i)} = \alpha_k\}|$ denote the number of labeled samples with the $k$th label.

**Theorem 4.1.** *Let $\widehat{\pi}_{\mathrm{MLE}}$ be the MLE defined in (4). If $\Delta_{\mathrm{MLE}} := \Delta_{\mathrm{MLE}}(\Lambda) > 0$ then*

$$\mathbb{P}(\widehat{\pi}_{\mathrm{MLE}} = \pi^*) \geq 1 - 2K^2 \exp\Big(-\inf_k n_k \cdot \inf_b \beta_b^*(\Delta_{\mathrm{MLE}}/3)\Big).$$

The condition $\Delta_{\mathrm{MLE}}(\Lambda) > 0$ is important to ensure that $\pi^*$ is learnable from $\Lambda$, and the size of $\Delta_{\mathrm{MLE}}(\Lambda)$ quantifies "how easy" it is to learn $\pi^*$ is given $\Lambda$. A bigger gap implies an easier problem. Thus, it is of interest to understand this quantity better. The following proposition shows that when $\Lambda = \Lambda^*$, this gap is always nonnegative:

**Proposition 4.2.** *For any permutation $\pi$ and any $\Lambda$,*

$$\mathbb{E}_*\ell(\pi;\Lambda,X,Y) \leq \mathbb{E}_*\ell(\pi^*;\Lambda^*,X,Y)$$

*and hence $\Delta_{\mathrm{MLE}}(\Lambda^*) \geq 0$.*

In general, assuming $\Delta_{\mathrm{MLE}}(\Lambda) > 0$ is a weak assumption, but bounds on $\Delta_{\mathrm{MLE}}(\Lambda)$ are difficult to obtain without making additional assumptions on the densities $f_k$ and $f_k^*$. A brief discussion of this can be found in Appendix B; we leave it to future work to study this quantity more carefully.

## 4.2 Majority vote

For any $\Lambda$, define $m_b := |i : X^{(i)} \in \mathcal{D}_b(\Lambda)|$ and $\chi_{bj}(\Lambda) := \frac{1}{m_b}\sum_{i=1}^{n} 1(Y^{(i)} = j, X^{(i)} \in \mathcal{D}_b(\Lambda))$, where $1(\cdot)$ is the indicator function. Similar to the MLE, our results for MV depend crucially on a "gap" quantity, given by

$$\Delta_{\mathrm{MV}}(\Lambda) := \inf_b \Big\{ \mathbb{E}_*\chi_{bb}(\Lambda) - \max_{j \neq b} \mathbb{E}_*\chi_{bj}(\Lambda) \Big\}. \tag{7}$$

This quantity essentially measures how much more likely it is to sample the $b$th label in the $b$th decision region than any other label, averaged over the entire region. Thus, conditions on $\Delta_{\mathrm{MV}}(\Lambda)$ are closely related to the well-known cluster assumption [4, 10, 11, 25].

**Theorem 4.3.** *Let $\widehat{\pi}_{\mathrm{MV}}$ be the MV defined in (5). If $\Delta_{\mathrm{MV}} := \Delta_{\mathrm{MV}}(\Lambda) > 0$ then*

$$\mathbb{P}(\widehat{\pi}_{\mathrm{MV}} = \pi^*) \geq 1 - 2K^2 \exp\Big(\frac{-2\Delta_{\mathrm{MV}}^2 \min_b m_b}{9}\Big).$$

As with the MLE, the gap $\Delta_{\mathrm{MV}}(\Lambda)$ is an important quantity. Fortunately, when $\Lambda = \Lambda^*$ it is always positive:

**Proposition 4.4.** *For each $b = 1, \ldots, K$,*

$$\mathbb{E}_*\chi_{bb}(\Lambda^*) > \max_{j \neq b} \mathbb{E}_*\chi_{bj}(\Lambda^*)$$

*and hence $\Delta_{\mathrm{MV}}(\Lambda^*) > 0$.*

When $\Lambda \neq \Lambda^*$, $\Delta_{\mathrm{MV}}(\Lambda)$ has the following interpretation: $\Delta_{\mathrm{MV}}(\Lambda)$ measures how well the decision regions defined by $\Lambda$ match up with the decision regions defined by $\Lambda^*$. When $\Lambda$ defines decision regions that assign high probability to one class, $\Delta_{\mathrm{MV}}(\Lambda)$ will be large. If $\Lambda$ defines decision regions where multiple classes have approximately the same probability, however, then it is possible that $\Delta_{\mathrm{MV}}(\Lambda)$ will be small. In this case, our experiments in Section 6 indicate that the MLE performs much better by managing overlapping decision regions more gracefully.

## 4.3 Sample complexity

Theorems 4.1 and 4.3 imply upper bounds on the minimum number of samples required to learn the permutation $\pi^*$: For any $\delta \in (0, 1)$, as long as

$$\text{(MLE)} \qquad \inf_k n_k := n_0 \geq \frac{\log \frac{2K^2}{\delta}}{\inf_b \beta_b^*(\Delta_{\text{MLE}}/3)} \qquad (8)$$

$$\text{(MV)} \qquad \inf_b m_b := m_0 \geq \frac{9 \log \frac{2K^2}{\delta}}{2\Delta_{\text{MV}}^2} \qquad (9)$$

we recover $\pi^*$ with probability at least $1 - \delta$. Surprisingly, as stated these lower bounds are dimension-free, however, in practice the gaps $\Delta_{\text{MLE}}$ and $\Delta_{\text{MV}}$ may be dimension-dependent.

To derive the sample complexity in terms of the total number of labeled samples $n$, it suffices to determine the minimum number of samples per class given $n$ draws from a multinomial random variable. For the general case with unequal probabilities, Lemma D.2 provides a precise answer. For simplicity here, we summarize the special case where each class (resp. decision region) is equally probable for the MLE (resp. MV).

**Corollary 4.5** (Sample complexity of MLE)**.** *Suppose that $\lambda_k^* = 1/K$ for each $k$, $\Delta_{\text{MLE}} > 0$, and*

$$n \geq K \log(K/\delta) \Big[ 1 + \frac{4}{\inf_b \beta_b^*(\Delta_{\text{MLE}}/3)} \Big].$$

*Then $\mathbb{P}(\widehat{\pi}_{\text{MLE}} = \pi^*) \geq 1 - \delta$.*

**Corollary 4.6** (Sample complexity of MV)**.** *Suppose that $\mathbb{P}(X \in \mathcal{D}_b(\Lambda)) = 1/K$ for each $k$, $\Delta_{\text{MV}} > 0$, and*

$$n \geq K \log(K/\delta) \Big[ 1 + \frac{18}{\Delta_{\text{MV}}^2} \Big].$$

*Then $\mathbb{P}(\widehat{\pi}_{\text{MV}} = \pi^*) \geq 1 - \delta$.*

**Coupon collector's problem and SSL** To better understand these bounds, consider arguably simplest possible case: Suppose that each density $f_k^*$ has disjoint support, $\lambda_k^* = 1/K$, and that we know $\Lambda^*$. Under these very strong assumptions, an alternative way to learn $\pi^*$ is to simply sample from $\mathbb{P}(X)$ until we have visited each decision region $\mathcal{D}_k^*$ at least once. This is the classical *coupon collector's problem* (CCP), which is known to require $\Theta(K \log K)$ samples [33, 34]. Thus, under these assumptions the expected number of samples required to learn $\pi^*$ is $\Theta(K \log K)$. By comparison, our results indicate that *even if the $f_k^*$ have overlapping supports* and *we do not know* $\Lambda^*$, as long as $\Delta_{\text{MLE}} = \Omega(1)$ (resp. $\Delta_{\text{MV}} = \Omega(1)$) then $\Omega(K \log K)$ samples suffice to learn $\pi^*$. In other words, SSL is approximately as difficult as CCP in very general settings.

## 4.4 Classification error

So far our results have focused on the probability of recovery of the unknown permutation $\pi^*$. We can further bound the classification error of the classifier (3) in terms of the Wasserstein distance $W_1(\Lambda, \Lambda^*)$ between $\Lambda$ and $\Lambda^*$ as follows:

**Theorem 4.7** (Classification error)**.** *Let $g^* = g_{\Lambda^*, \pi^*}$ denote the Bayes classifier. If $\pi^*(\alpha_b) = \arg\min_i d_{\text{TV}}(f_i, f_b^*)$ then there is a constant $C > 0$ depending on $K$ and $\Lambda^*$ such that*

$$\mathbb{P}(g_{\Lambda, \pi^*}(X) \neq Y) \leq \mathbb{P}(g^*(X) \neq Y) + C \cdot W_1(\Lambda, \Lambda^*) + \sum_b |\lambda_{\pi^*(\alpha_b)} - \lambda_b^*|.$$

This theorem allows for the possibility that the mixture model $\Lambda$ learned from the unlabeled data is not the same as $\Lambda^*$ (e.g. the true mixing measure corresponding to the true class conditionals). It is thus necessary to assume that the mismatch between $\Lambda$ and $\Lambda^*$ is not so bad that the closest density $f_i$ to $f_b^*$ is something other than $f_{\pi^*(\alpha_b)}$.

The interpretation of this theorem is as follows: Given $\Lambda$, we learn a permutation $\widehat{\pi} = \widehat{\pi}_n(\Lambda)$ from $n$ labeled samples, e.g. using either the MLE (4) or MV (5). Together, the pair $(\Lambda, \widehat{\pi})$ defines a classifier

$g_{\Lambda,\widehat{\pi}}$ via (3). We are interested in bounding the probability of misclassification $\mathbb{P}(g_{\Lambda,\widehat{\pi}}(X) \neq Y)$ in terms of the Bayes error. Since $\widehat{\pi} = \pi^*$ with high probability, Theorem 4.7 implies that

$$\mathbb{P}(g_{\Lambda,\widehat{\pi}}(X) \neq Y) \leq \mathbb{P}(g^*(X) \neq Y) + C \cdot W_1(\Lambda, \Lambda^*) + \sum_b |\lambda_{\pi^*(\alpha_b)} - \lambda_b^*|.$$

In this case, there is an irreducible error quantified by the Wasserstein distance $W_1(\Lambda, \Lambda^*)$. In fact, if $W_1(\Lambda, \Lambda^*) = 0$, then Theorem 4.7 implies that $\mathbb{P}(g_{\Lambda,\widehat{\pi}}(X) \neq Y) \leq \mathbb{P}(g^*(X) \neq Y)$, i.e. the excess risk is zero. This is clearly a very strong conclusion.

**Identifiability and misspecification**  As discussed in Section 2, although $\Lambda^*$ will in general be nonidentifiable, the unlabeled data may identify some other mixing measure $\Lambda$ (see e.g. [17]). Suppose that $\widehat{\Lambda}_m$ is a mixing measure estimated from $m$ unlabeled samples and that $W_1(\widehat{\Lambda}_m, \Lambda) \to 0$. The question then is how much the misspecified $\Lambda$ helps in classification.

**Corollary 4.8.** *Suppose $W_1(\widehat{\Lambda}_m, \Lambda) = O(r_m)$ for some $r_m \to 0$ where $m$ is the number of unlabeled samples and $\pi^*(\alpha_b) = \arg\min_i d_{\mathrm{TV}}(f_i, f_b^*)$. Then if $\widehat{\pi} = \pi^*$,*

$$\mathbb{P}(g_{\widehat{\Lambda}_m,\widehat{\pi}}(X) \neq Y) \leq \mathbb{P}(g^*(X) \neq Y) + C \cdot r_m + C \cdot W_1(\Lambda, \Lambda^*).$$

In particular, if $W_1(\Lambda, \Lambda^*) = 0$, then

$$\mathbb{P}(g_{\widehat{\Lambda}_m,\widehat{\pi}}(X) \neq Y) - \mathbb{P}(g^*(X) \neq Y) = O(r_m).$$

**Clairvoyant SSL**  Previous work [5, 6, 11] has studied the so-called *clairvoyant* SSL case in which it is assumed that we know (1) perfectly. This amounts to taking $\widehat{\Lambda}_m = \Lambda$ in the previous results, or equivalently $m = \infty$. Under this assumption, we have perfect knowledge of the decision regions and only need to learn the label permutation $\pi^*$. Then Corollary 4.8 implies that with high probability, we can learn a Bayes classifier for the problem using finitely many labeled samples.

**Convergence rates**  The convergence rate $r_m$ used here is essentially the rate of convergence in estimating an identifiable mixture model, which is well-studied for parametric mixture models [35–37]. In particular, for so-called *strongly* identifiable parametric mixture models, the minimax rate of convergence attains the optimal root-$m$ rate $r_m = m^{-1/2}$ [35].[1] Asymptotic consistency theorems for nonparametric mixtures can be found in Aragam et al. [17].

**Comparison to supervised learning (SL).**  Previous work [11] has compared the sample complexity of SSL to SL under a cluster-type assumption. While a precise characterization of these trade-offs is not the main focus of this paper, we note in passing here the following: If the minimax risk of SL for a particular problem is larger than $W_1(\Lambda, \Lambda^*)$, then Theorem 4.7 implies that SSL provably outperforms SL on finite samples.

## 5   Algorithms

One of the significant appeals of MV (5) is its simplicity. It is conceptually easy to understand and trivial to implement. The MLE (4), on the other hand, is more subtle and difficult to compute in practice. In this section, we discuss two algorithms for computing the MLE: 1) An exact algorithm based on finding the maximum weight perfect matching in a bipartite graph by the Hungarian algorithm [39], and 2) Greedy optimization.

Define $C_k = \{i : Y^{(i)} = \alpha_k\}$. Consider the weighted complete bipartite graph $G = (V_{K,K}, w)$ with edge weights

$$w(k, k') = \sum_{i \in C_k} \log\left(\lambda_{k'} f_{k'}(X^{(i)})\right), \quad \forall k, k' \in [K]$$

Since a permutation $\pi$ defines a perfect matching on $G$, the log-likelihood can be rewritten as

$$\ell_n(\pi; \Lambda) = \sum_{k=1}^{K} \sum_{i \in C_k} \log\left(\lambda_{\pi(\alpha_k)} f_{\pi(\alpha_k)}(X^{(i)})\right) = \sum_{k=1}^{K} w(k, \pi(\alpha_k)),$$

the right side of which is the total weight of the matching $\pi$. Hence, the maximizer $\widehat{\pi}_{\text{MLE}}$ can be found by finding a perfect matching for this graph that has maximum weight. This can be done in $O(K^3)$ using the well-known Hungarian algorithm [39].

We can also approximately solve the matching problem by a greedy method: Assign the $k$th class to

$$\widehat{\pi}_{\text{G}}(\alpha_k) = \underset{k' \in [K]}{\arg\max} \, w(k, k') = \underset{k' \in [K]}{\arg\max} \sum_{i \in C_k} \log \left( \lambda_{k'} f_{k'}(X^{(i)}) \right),$$

This greedy heuristic isn't guaranteed to achieve optimal matching, however, it is simple to implement and can be viewed as a "soft interpolation" of $\widehat{\pi}_{\text{MLE}}$ and $\widehat{\pi}_{\text{MV}}$ as follows: If we define $w_{\text{MV}}(k, k') = \sum_{i \in C_k} 1(X^{(i)} \in \mathcal{D}_{k'}(\Lambda))$, we can see that a training example $(X^{(i)}, Y^{(i)} = \alpha_k)$ contributes 1 to $w_{\text{MV}}(k, k')$ if $k' = \arg\max_j \lambda_j f_j(X^{(i)})$, and contributes 0 to $w_{\text{MV}}(k, k')$ otherwise. By comparison, for the greedy heuristic, a training example $(X^{(i)}, Y^{(i)} = \alpha_k)$ contributes $\log(\lambda_{k'} f_{k'}(X^{(i)}))$ to $w(k, k')$. Therefore, the greedy estimator can be seen as a "soft" version of MV that also greedily optimizes the MLE objective.

## 6 Experiments

In order to evaluate the relative performance of the proposed estimators in practice, we implemented each of the three methods described in Section 5 on simulated and real data. These experiments also illustrate the gap $\Delta_{\text{MLE}}(\Lambda)$ (resp. $\Delta_{\text{MV}}(\Lambda)$) that appears in Theorem 4.1 (resp. Theorem 4.3): In many of the examples, although the learned mixture is badly misspecified and the true class conditionals overlap significantly, it is still possible to recover $\pi^*$ with fewer than 100 labeled samples (sometimes significantly fewer).

Our experiments consider three settings: (i) Parametric mixtures of Gaussians, (ii) A nonparametric mixture model, and (iii) Real data from MNIST. In each experiment, a random true mixture model $\Lambda^*$ was generated from one of these settings, and then $N = 99$ labeled samples were drawn from this mixture model. We generated $\Lambda^*$ under different separation conditions, from well-separated to overlapping. Then, $\Lambda$ was generated in two ways: (a) $\Lambda = \Lambda^*$, corresponding to a setting where the true decision boundaries are known, and (b) $\Lambda \neq \Lambda^*$ by perturbing the components and weights of $\Lambda^*$ by a parameter $\eta > 0$ (see Appendix A for details). $\Lambda$ was then used to estimate $\pi^*$ using each of the three algorithms described in the previous section for the first $n = 3, 6, 9, \ldots, 99$ labeled samples. This procedure was repeated $T = 50$ times (holding $\Lambda^*$ and $\Lambda$ fixed) in order to estimate $\mathbb{P}(\widehat{\pi} = \pi^*)$. Full details of the experiments can be found in Appendix A.

**Mixture of Gaussians** A random Gaussian mixture model with $K \in \{2, 4, 9, 16\}$ and dimension $d = 2$. Since the lower bounds (8) and (9) are dimension-free, we tested examples with $d = 10$ as well with similar results.

**Nonparametric mixture model** A nonparametric mixture model with $K = 4$. Each $f_k^*$ was chosen to be a random Gaussian mixture. Thus, the overall density is a "mixture of Gaussian mixtures".

**MNIST** To approximate real data, we used training data from the MNIST dataset to build $K = 10$ class conditionals $f_k^*$ from real data using kernel density estimates. For labeled data, we sampled from the test data. To simulate the case $\Lambda \neq \Lambda^*$, we contaminated the training labels by randomly switching 10% of the labels.

The results are shown in Figure 2. As expected, the MLE performs by far the best, obtaining near perfect recovery of $\pi^*$ with fewer than $n = 20$ labeled samples on synthetic data, and fewer than $n = 40$ on MNIST. Unsurprisingly, the most difficult case was $K = 16$, in which only the MLE was able recover the true permutation $> 50\%$ of the time. By increasing $n$, the MLE is eventually able to learn this most difficult case, in accordance with our theory. Furthermore, the MLE is much more robust to misspecification $\Lambda \neq \Lambda^*$ and component overlap compared to the others. This highlights the advantage of leveraging density information in the MLE, which is ignored by the MV estimator). These results also illustrate how the gaps $\Delta_{\text{MLE}}$ and $\Delta_{\text{MV}}$ affect learning $\pi^*$: Even when $W_1(\Lambda, \Lambda^*)$ is large, the labeled sample complexity is relatively small (fewer than $n = 100$ in general). For example, see Fig 4 in Appendix A for an illustration of the difficulty of the case $K = 16$. Furthermore, in Appendix A, we also compare the classification accuracy of the resulting SSL classifiers with a

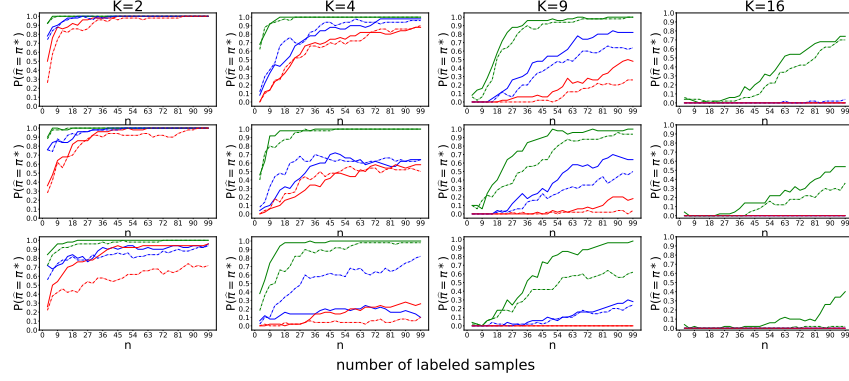

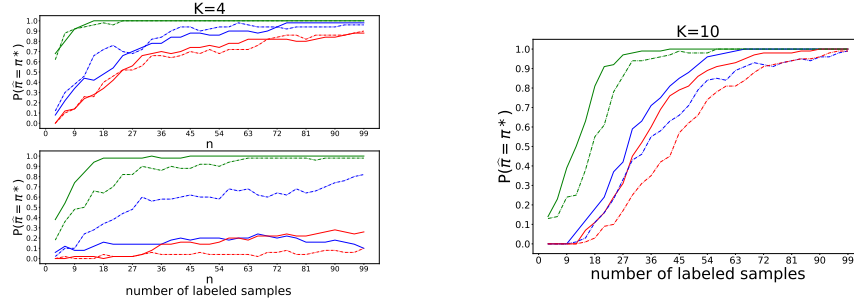

(a) Mixture of Gaussians

(b) Mixture of Gaussian mixtures

(c) MNIST

Figure 2: Performance of MLE (Hungarian - Green; Greedy - Blue) and MV (Red). Solid line and dashed line correspond to the performance when $\Lambda^* = \Lambda$ and $\Lambda^* \neq \Lambda$, respectively. Columns correspond to the number of classes $K$; rows correspond to decreasing separation; e.g. the bottom rows in each figure are the least separated.

standard supervised baseline (LeNet) on the MNIST dataset. When sample size is small, it is clear that our proposed estimators are more accurate. In accordance with our theory, the accuracy of the SSL classifiers plateaus around 96% due to misspecification of $\Lambda^*$, as measured by $W_1(\Lambda, \Lambda^*)$.

# 7   Discussion

Using nonparametric mixture models as a foundation, we analyzed the labeled sample complexity of semi-supervised learning. Our results allow for arbitrary, possibly heuristic estimators of a mixing measure $\Lambda$ that is used to approximate the unlabeled data distribution $F^*$. This mixing measure defines decision boundaries that can be used to define a semi-supervised classifier whose classification accuracy is controlled by the Wasserstein distance between $\Lambda$ and $\Lambda^*$, the true mixing measure corresponding to the class conditional distributions. This draws an explicit connection between the quality of what is learned from the unlabeled data (i.e. $\Lambda$) and the quality of the resulting classifier. Our experiments convey two main takeaway messages: 1) It pays off to use density information as with the MLE, and 2) When the mixture model learned from the unlabeled data is a poor approximation of the true mixing measure, or the true class conditionals have substantial overlap, the MLE can still learn a reasonable semi-supervised classifier.

This work poses many interesting questions for future work, including instantiating our results for practical methods for learning $\Lambda$ and quantifying the dependence of $\Delta_{\text{MLE}}$ and $\Delta_{\text{MV}}$ on $K$ and $d$. Furthermore, it would be interesting to provide a rigorous comparison of $\Delta_{\text{MLE}}$ and $\Delta_{\text{MV}}$ in specific settings in order to better understand the trade-off between the MLE and MV estimators. Finally, exploring additional connections with existing assumptions such as the cluster and manifold assumptions is an interesting problem.

**Acknowledgments**

P.R. acknowledges the support of NSF via IIS-1149803, IIS-1664720, DMS-1264033, and ONR via N000141812861. E.X. acknowledges the support of NIH R01GM114311, P30DA035778.

## Footnotes

[1]This paper corrects an earlier result due to Chen [38] that claimed an $m^{-1/4}$ minimax rate.

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
