[Supplementary Material]

# A  Experiment details

We evaluate the performance of the maximum likelihood estimator and majority vote estimator that are described in Section 3. For maximum likelihood estimator, we have implemented both the *Hungarian Algorithm* and *Greedy Algorithm.*

Some of the examples used were quite difficult: See Figure 4.

**Mixture of Gaussians**    We begin our experiment with synthetic data where $F^* = \sum_k \lambda_k^* f_k^*$ is a mixture of Gaussian with $\lambda_k^*$ being randomly drawn from a uniform distribution $\mathcal{U}(0,1)$ (normalized afterwards) and $f_k^*$ being a Gaussian density. The $f_k^*$ were arranged on a square grid with randomly generated PSD covariance matrices and then normalized by multiplying by a random scale factor $a \sim \mathcal{U}(0.3, 0.6)$.

There are three main parameters of interest in the experiments: 1) The number of clusters $K$, 2) The degree of separation among the Gaussians, and 3) The dimension of the mixture model.

To explicitly control how well-separated the Gaussians are, we shrink the expectations of the Gaussians towards the origin using a parameter $\eta$ where $\eta$ can be $1, 0.75, 0.5$. We design the means of the Gaussians so that they are on a grid centered at the origin. The mean of the Gaussian is thus given by $\eta \times \mu_k^*$, where $\mu_k^*$ is the mean of the $k$th density. When $\eta = 1$, components in the mixture are well-separated where $\{f_k^*\}_{k=1}^K$ have no or very little overlap within one standard deviation. The smaller the $\eta$ is, the more cluttered the components are. For each choice of dimension (either 2 or 10), $K$ can be $\{2, 4, 9, 16\}$.

We first sample 50 sets of samples with each set of size $n = 99$. Then, we evaluate the algorithms at $n = 3, 6, ..., 99$. To make sure that the performance of the three algorithms at different sample size is comparable, we add 3 labeled samples every time when the sample size is incremented. Thus at each sample size $n$, the algorithms are tested on the 50 different datasets.

**Perturbed mixture of Gaussian**    In this setting, we test the case where $\Lambda^*$ is unknown and the algorithms only have access to its perturbed version $\Lambda$. Similar to the above setups, we sample $n$ labeled data using $\Lambda^*$. However, instead of feeding the algorithms with $n$ labeled samples and $\Lambda^*$, we give the algorithm $\Lambda$ where mixture weights are shifted: Each dimension of the means of the Gaussians are shifted by a random number drawn from $\mathcal{N}(0, 0.1)$ and the variance of each Gaussians is scaled by either $0.5$ or $2$ (chosen at random).

**Mixture of Gaussian mixtures and its perturbation**    The experiment of the mixture of Gaussian mixtures is similar to the one of mixture of Gaussian. In this case, each $f_k^*$ is a Gaussian mixture. We also controlled the separation condition by shrinking the expectation of each Gaussian towards the origin where $\eta = 1$ or $0.5$.

**MNIST and corrupted MNIST**    We train 10 kernel density estimator (one per digit) with bandwidth being 1 for $\{f_k\}_{k=1}^{10}$ using $60,000$ training data points. The algorithms are then tested using the unseen test data and the learned densities $\{f_k\}_{k=1}^{10}$. (Note that the density estimation in this case may not be perfect but because each cluster is well-separated, both MLE and majority vote estimator still perform well.)

We then test, under corruption of the labeled samples from the test set, how the three algorithms behave. With probability $0.1$, the label of the sampled data is changed to an incorrect label.

**Classification Accuracy**    We also compared the classification accuracy of our proposed SSL estimators (MLE and MV) with a canonical supervised baseline for MNIST, the LeNet convolutional neural network. Similar as before, we used the training set in the MNIST dataset as our unlabeled samples to learn the mixture model by training a kernel density estimator per digit. We randomly sampled 99 labeled samples as our training data for the SSL estimators and LeNet. We then tested the learned classifier on $1,000$ random test points. As seen in Figure 3, our estimators attained higher accuracy faster with fewer samples. It is also interesting to see that the SSL estimators stopped increasing their performance once the accuracy reached $\approx .962$. This is due to the fact that the kernel density estimation was not perfect. And as our theory suggested, the performance plateaus due to the gap $W_1(\Lambda, \Lambda^*)$.

(a)

(b)

Figure 3: (a) shows the performance of MLE (Hungarian - Green; Greedy - Blue) and MV (Red) when $K = 16$ for $n > 99$. The three rows corresponds to $\eta = 1, 0.75, 0.5$. In (b), we show the classification accuracy of the three estimators on MNIST and LeNet (Yellow) trained on the labeled samples. In both cases, the solid line and dashed line correspond to the performance when $\Lambda^* = \Lambda$ and $\Lambda^* \neq \Lambda$, respectively.

Figure 4: Some examples used in the experiments. Depicted are contour lines of the densities for one standard deviation from the mean. (top) Mixture of Gaussians with $K = 16$. (bottom) Nonparametric mixture of Gaussian mixtures; each Gaussian component is coloured according to the class label it generates.

# B Discussion of conditions

Here we have a simple experiment with the underlying distribution being a mixture of two Gaussians:

$$F = \frac{1}{2}\lambda_1^* + \frac{1}{2}\lambda_2^* = \frac{1}{2}\mathcal{N}(-\mu, 1) + \frac{1}{2}\mathcal{N}(\mu, 1)$$

where $\mu$ is a small positive number indicating the separation between two Gaussians. We would like to compare the number of samples needed to recover the true permutation $\pi^*$ with probability $(1 - \delta)$ for both MLE and MV.

Our experiments show that both estimators have roughly $O(\mu^{-2})$ sample complexity when $\mu \to 0^+$, but MV needs about **4 times** as many samples as the MLE. In fact, our theory can verify the sample complexity of MV: The gap $\Delta_{\mathrm{MV}}$ is $\Phi(\mu) - \Phi(-\mu) = O(\mu)$ and the sample complexity has $\log(K/\delta)/\Delta_{\mathrm{MV}}^2$ dependence with $\Delta_{\mathrm{MV}}$, which gives exactly $O(\mu^{-2})$. Here $\Phi(\mu)$ is the cumulative distribution function of standard normal random variable. Unfortunately, the intractable form of the dual functions $\beta_b^*$ makes similar analytical comparisons difficult.

# C Proofs

## C.1 Proof of Theorem 4.1

*Proof.* Denote a maximizer of the expected log-likelihood by $\widetilde{\pi} \in \arg\max \mathbb{E}_* \ell(\pi; \Lambda)$ and define $\Delta(\pi) = \mathbb{E}_* \ell(\widetilde{\pi}; \Lambda, X, Y) - \mathbb{E}_* \ell(\pi; \Lambda, X, Y)$. Note that $\Delta(\pi) \geq \Delta > 0$ for all $\pi \neq \widetilde{\pi}$. Define $\mathcal{A}_\pi(t) = \{|\ell(\pi; \Lambda, X, Y) - \mathbb{E}_* \ell(\pi; \Lambda, X, Y))| < t\}$.

Then for any $t < \Delta/2 \leq \Delta(\pi)/2$, on the event $\cap_\pi \mathcal{A}_\pi(t)$ we have

$$
\begin{aligned}
\ell(\widetilde{\pi}; \Lambda, X, Y) &> \mathbb{E}_* \ell(\widetilde{\pi}; \Lambda, X, Y) - t \\
&> \mathbb{E}_* \ell(\pi; \Lambda, X, Y) + \Delta(\pi) - 2t \\
&> \ell(\pi; \Lambda, X, Y) \quad \forall \pi \neq \widetilde{\pi}.
\end{aligned}
$$

Invoking Lemma D.1 with $g_k(X, Y) = \log \lambda_k f_k(X, Y)$, we have

$$
\begin{aligned}
\mathbb{P}(\cap_\pi \mathcal{A}_\pi(t)) &= \mathbb{P}\Big(\forall \pi, \Big|\frac{1}{n}\sum_{i=1}^n \ell(\widetilde{\pi}; \Lambda, X^{(i)}, Y^{(i)}) - \mathbb{E}_* \ell(\widetilde{\pi}; \Lambda, X^{(i)}, Y^{(i)})\Big| \leq t\Big) \\
&\geq 1 - 2K^2 \exp(-\inf_k \inf_b n_k \beta_b^*(t))
\end{aligned}
$$

Therefore, making the arbitrary choice of $t = \Delta/3$,

$$
\begin{aligned}
\mathbb{P}(\widehat{\pi} = \widetilde{\pi}) &= \mathbb{P}\big(\ell(\widetilde{\pi}; \Lambda, X, Y) > \ell(\pi; \Lambda, X, Y) \, \forall \pi \neq \widetilde{\pi}\big) \\
&\geq 1 - 2K^2 \exp(-\inf_k \inf_b n_k \beta_b^*(\Delta/3)).
\end{aligned}
$$

Since $\Delta > 0 \implies \pi^* = \widetilde{\pi}$, the desired result follows. $\qquad\square$

## C.2 Proof of Proposition 4.2

*Proof.* Let $p(x, y) = \lambda_{\pi^*(y)}^* f_{\pi^*(y)}^*(x)$, $q(x, y) = \lambda_{\pi(y)} f_{\pi(y)}(x)$, so that

$$
\begin{aligned}
\mathbb{E}_* \ell(\pi^*; \Lambda^*, X, Y) - \mathbb{E}_* \ell(\pi; \Lambda, X, Y) &= \mathbb{E}_* \log(p(x, y)) - \mathbb{E}_* \log(q(x, y)) \\
&= \int_x \sum_y p(x, y) \log \frac{p(x, y)}{q(x, y)} dx \\
&= \mathrm{KL}(p \,||\, q) \\
&\geq 0.
\end{aligned}
$$

The equality holds if and only if $p(x, y) = q(x, y)$ holds for all $x, y$. $\qquad\square$

## C.3 Proof of Theorem 4.3

*Proof.* We have

$$\mathbb{P}(\widehat{\pi} = \pi) = \mathbb{P}\big(\underbrace{\widehat{\pi}(b) = b}_{\mathcal{E}_b} \ \forall b \in [K]\big) = \mathbb{P}\Big(\bigcap_{b=1}^{K} \mathcal{E}_b\Big),$$

where

$$\mathcal{E}_b = \left\{ \sum_{i=1}^{n} 1(Y^{(i)} = b, X^{(i)} \in \mathcal{D}_b(\Lambda)) > \sum_{i=1}^{n} 1(Y^{(i)} = j, X^{(i)} \in \mathcal{D}_b(\Lambda)) \quad \forall j \neq b \right\}.$$

Let $U_{bj}^{(i)} := 1(Y^{(i)} = j, X^{(i)} \in \mathcal{D}_b(\Lambda))$ so that $\chi_{bj} = \frac{1}{n_b} \sum_i U_{bj}^{(i)}$. It suffices to control the event

$$\left\{ \sum_{i=1}^{n} U_{bb}^{(i)} > \sum_{i=1}^{n} U_{bj}^{(i)} \quad \forall j \neq b \right\} = \{\chi_{bb} > \chi_{bj} \ \forall j \neq b\} \tag{10}$$

where $U_j^{(i)} \in \{0, 1\}$ are i.i.d. random variables. Thus, we are interested in the probability $\mathbb{P}(\chi_{bb} > \chi_{bj} \ \forall j \neq b)$. Note that

$$\mathbb{E}_* \chi_{bj} = \frac{1}{n_b} \sum_{i=1}^{n} \mathbb{E}_* U_{bj}^{(i)} = \frac{1}{n_b} \sum_{i:X^{(i)} \in \mathcal{D}_b} \mathbb{P}(Y^{(i)} = j, X^{(i)} \in \mathcal{D}_b(\Lambda)).$$

Define

$$\Delta_{bj} := \mathbb{E}_* \chi_{bb} - \mathbb{E}_* \chi_{bj} \tag{11}$$

and $\mathcal{A}_{bj}(t) = \{|\chi_{bj} - \mathbb{E}_* \chi_{bj}| < t\}$. Then for any $t < \Delta/2$, on the event $\cap_{j=1}^{K} \mathcal{A}_{bj}(t)$ we have

$$\chi_{bb} > \mathbb{E}_* \chi_{bb} - t > \mathbb{E}_* \chi_{bj} + \Delta - 2t > \chi_{bj} \quad \forall j \neq b.$$

In other words, making the arbitrary choice of $t = \Delta/3$, we deduce

$$\mathbb{P}\big(\mathcal{E}_b^c\big) \leq \mathbb{P}\Big(\bigcup_{j=1}^{K} \mathcal{A}_j(\Delta/3)^c\Big) \leq 2K \exp(-2n_b \Delta^2/9)$$

where we used Hoeffding's inequality to bound $\mathbb{P}\big(\mathcal{A}_j(\Delta/3)^c\big)$ for each $j$.

Thus

$$\begin{aligned}
\mathbb{P}\Big(\bigcap_{b=1}^{K} \mathcal{E}_b\Big) &= 1 - \sum_{b=1}^{K} \mathbb{P}\Big(\bigcup_{j=1}^{K} \mathcal{A}_j(\Delta/3)^c\Big) \\
&\geq 1 - 2K \sum_{b=1}^{K} \exp(-2n_b \Delta^2/9) \\
&\geq 1 - 2K^2 \exp\Big(\frac{-2\Delta^2 \min_b n_b}{9}\Big),
\end{aligned}$$

as claimed. $\qquad\square$

## C.4 Proof of Proposition 4.4

*Proof.* We have for any $j \neq b$,

$$
\begin{aligned}
\mathbb{E}_* \chi_{bb}(\Lambda^*) &= \frac{1}{n_b} \sum_{i=1}^{n} \mathbb{E}_* 1(Y^{(i)} = b, X^{(i)} \in \mathcal{D}_b(\Lambda)) \\
&= \frac{1}{n_b} \sum_{i=1}^{n} \mathbb{P}(Y^{(i)} = b, X^{(i)} \in \mathcal{D}_b(\Lambda)) \\
&= \frac{1}{n_b} \sum_{i=1}^{n} \mathbb{P}(Y^{(i)} = b \mid X^{(i)} \in \mathcal{D}_b(\Lambda)) \mathbb{P}(X^{(i)} \in \mathcal{D}_b(\Lambda)) \\
&> \frac{1}{n_b} \sum_{i=1}^{n} \mathbb{P}(Y^{(i)} = j \mid X^{(i)} \in \mathcal{D}_b(\Lambda)) \mathbb{P}(X^{(i)} \in \mathcal{D}_b(\Lambda)) \\
&= \mathbb{E}_* \chi_{bj}(\Lambda^*). \qquad \square
\end{aligned}
$$

## C.5 Proof of Corollaries 4.5 and 4.6

We prove Corollary 4.5; the proof of Corollary 4.6 is similar with $n_k$ replaced by $m_b$ and $n_0$ in (8) by $m_0$ in (9).

*Proof.* Using $p_k = 1/K$ in Lemma D.2, we deduce for any $m > 0$

$$
\mathbb{P}(\min_k n_k \geq m) \geq 1 - K \exp\left(-\frac{2K}{n}(n/K - m)^2\right).
$$

Thus, for any $\delta > 0$, we have

$$
n \geq \frac{K}{2}\left[\log(K/\delta) + 4m\right] \implies \mathbb{P}(\min_k n_k \geq m) \geq 1 - \delta.
$$

The desired result follows from replacing $m$ with the lower bound on $n_0$ in (8) and invoking Theorem 4.1. $\qquad \square$

## C.6 Proof of Theorem 4.7

The following lemma—used in the proof of Theorem 4.7—is also useful in case the main assumption of Theorem 4.7 is violated.

**Lemma C.1.** *Assume without loss of generality that $\pi^*(\alpha_k) = k$. Then for any $\Lambda$, the classification error of $g_{\Lambda, \pi^*}$ can be bounded as follows:*

$$
\mathbb{P}(g_{\Lambda, \pi^*}(X) \neq Y) \leq \mathbb{P}(g^*(X) \neq Y) + \sum_b \mathbb{P}(X \in \mathcal{D}_b \triangle \mathcal{D}_b^*).
$$

*Proof.* See, e.g. §2.5 in Devroye et al. [40] for the case $K = 2$, the general case $K > 2$ is proved similarly. $\qquad \square$

*Proof of Theorem 4.7.* Write $f_k$ for the components of $\Lambda$ and $\lambda_k$ for the corresponding weights. Without loss of generality, assume $\pi^*(\alpha_k) = k$, so that $\mathcal{D}_b(\Lambda)$ corresponds to the decision region for the class $\alpha_b$. Then using Lemma C.1 we have

$$
\begin{aligned}
\mathbb{P}(g_{\Lambda, \pi^*}(X) \neq Y) &\leq \mathbb{P}(g^*(X) \neq Y) + \sum_b \mathbb{P}(X \in \mathcal{D}_b \triangle \mathcal{D}_b^*) \\
&\leq \mathbb{P}(g^*(X) \neq Y) + \sum_b \int_{\mathcal{X}} |\lambda_b f_b(x) - \lambda_b^* f_b^*(x)| \, dx \\
&\leq \mathbb{P}(g^*(X) \neq Y) + \sum_b \left(|\lambda_b - \lambda_b^*| + \lambda_b^* \, d_{\mathrm{TV}}(f_b, f_b^*)\right) \\
&\leq \mathbb{P}(g^*(X) \neq Y) + C(\Lambda^*) \cdot W_1(\Lambda, \Lambda^*) + \sum_b |\lambda_b - \lambda_b^*|,
\end{aligned}
$$

where we invoked Lemma D.3 in the last line. □

# D  Additional lemmas

## D.1  Lemma D.1

For ease of notation in the following lemma, assume without loss of generality that $Y \in [K]$.

**Lemma D.1.** *Let* $g_1, \ldots, g_K$ *be functions and* $\psi_k(s) = \log \mathbb{E}_* \exp(s g_k(X, Y))$ *be the log moment generating function of* $g_k(X, Y)$. *Then*

$$\mathbb{P}\Big(\forall \pi : \frac{1}{n} \sum_{i=1}^{n} g_{\pi(Y_i)}(X_i) - \mathbb{E} g_{\pi(Y_i)}(X_i) \leq t\Big) \geq 1 - K^2 \exp(- \inf_k \inf_b n_k \psi_b^*(t)).$$

*Proof.* Define $C_k := \{i : Y_i = k\}$, $n_k := |C_k|$, and note that

$$\{i : \pi(Y_i) = b\} = \{i : Y_i = \pi^{-1}(b)\} = C_{\pi^{-1}(b)}.$$

Then we have the following:

$$Z := \frac{1}{n} \sum_{i=1}^{n} g_{\pi(Y_i)}(X_i) - \mathbb{E} g_{\pi(Y_i)}(X_i) = \frac{1}{n} \sum_{k=1}^{K} \sum_{i : \pi(Y_i) = b} g_b(X_i) - \mathbb{E} g_b(X_i)$$

$$= \frac{1}{n} \sum_{b=1}^{K} \sum_{i \in C_{\pi^{-1}(b)}} g_b(X_i) - \mathbb{E} g_b(X_i)$$

$$= \frac{1}{n} \sum_{b=1}^{K} n_{\pi^{-1}(b)} \underbrace{\Big\{ \frac{1}{n_{\pi^{-1}(b)}} \sum_{i \in C_{\pi^{-1}(b)}} g_b(X_i) - \mathbb{E} g_b(X_i) \Big\}}_{:= \widetilde{Z}_b(\pi)}$$

$$= \sum_{b=1}^{K} \frac{n_b(\pi)}{n} \widetilde{Z}_b(\pi).$$

Now, for each $\pi$, $\widetilde{Z}_b(\pi)$ is just a sum over one of $K$ possible subsets of $[n]$, i.e. samples indices. To see this, define

$$Z_{b,k} := \frac{1}{n_k} \sum_{i \in C_k} g_b(X_i) - \mathbb{E} g_b(X_i)$$

and note that $\widetilde{Z}_b(\pi) = Z_{b, \pi^{-1}(b)}$ for each $b$. It follows that

$$Z = \sum_{b=1}^{K} \frac{n_b(\pi)}{n} \widetilde{Z}_b(\pi) = \sum_{b=1}^{K} \frac{n_{\pi^{-1}(b)}}{n} Z_{b, \pi^{-1}(b)}$$

Chernoff's inequality implies $\mathbb{P}(Z_{b,k} \geq t) \leq \exp(-n_k \psi_b^*(t))$ for each $b$ and $k$, which implies that

$$\mathbb{P}(\sup_{b,k} Z_{b,k} < t) = \mathbb{P}\Big(\bigcap_k \bigcap_b \{Z_{b,k} < t\}\Big)$$

$$\geq 1 - \mathbb{P}\Big(\bigcup_k \bigcup_b \{Z_{b,k} < t\}^c\Big)$$

$$\geq 1 - \sum_{k=1}^{K} \sum_{b=1}^{K} \mathbb{P}(Z_{b,k} \geq t)$$

$$\geq 1 - \sum_{k=1}^{K} \sum_{b=1}^{K} \exp(-n_k \psi_b^*(t))$$

$$\geq 1 - K^2 \exp(- \inf_k \inf_b n_k \psi_b^*(t)).$$

Now, if $\sup_{b,k} Z_{b,k} < t$, then

$$Z = \sum_{b=1}^{K} \frac{n_{\pi^{-1}(b)}}{n} Z_{b,\pi^{-1}(b)} < \sum_{b=1}^{K} \frac{n_{\pi^{-1}(b)}}{n} t = t$$

since $\sum_b n_b/n = 1$ and $\pi$ is a bijection. The desired result follows. $\qquad\square$

## D.2   Lemma D.2

The following lemma gives a precise bound on the minimum number of samples $n$ required to ensure $\min_k n_k \geq m$ from a generic multinomial sample with high probability:

**Lemma D.2.** *Let $Y_i$ be a multinomial random variable such that $\mathbb{P}(Y_i = k) = p_k$ and define $n_k = \sum_{i=1}^{n} 1(Y_i = k)$. Then for any $m > 0$,*

$$\mathbb{P}(\min_k n_k \geq m) \geq 1 - \sum_{k=1}^{K} \exp\left(-\frac{2}{np_k}(np_k - m)^2\right).$$

*Proof.* By standard tail bounds on $n_k \sim \text{Bin}(n, p_k)$, we have $\mathbb{P}(n_k \leq m) \leq \exp(-2(np_k - m)^2/(np_k))$. Thus

$$\mathbb{P}(\min_k n_k < m) = \mathbb{P}(\cup_{k=1}^{K}\{n_k < m\}) \leq \sum_{k=1}^{K} \mathbb{P}(n_k < m) \leq \sum_{k=1}^{K} \exp\left(-\frac{2}{np_k}(np_k - m)^2\right),$$

as claimed. $\qquad\square$

## D.3   Lemma D.3

For any density $f \in L^1$, let $\delta_f$ denote the point mass concentrated at $f$, so that for any Borel subset $A \subset \mathcal{P}$,

$$\delta_f(A) = \begin{cases} 1, & f \in A \\ 0, & f \notin A. \end{cases}$$

The following lemma is standard and hence the proof is omitted:

**Lemma D.3.** *Let $\Lambda = \sum_{k=1}^{K} \lambda_k \delta_{f_k}$ and $\Lambda' = \sum_{k=1}^{K} \lambda'_k \delta_{f'_k}$. Then there is a constant $C = C(\Lambda', K)$ such that*

$$\sup_j \inf_i \ d_{\text{TV}}(f_i, f'_j) \leq C \, W_1(\Lambda, \Lambda'). \tag{12}$$