[Reviews · NeurIPS 2018]

Reviewer 1



# Summary The authors try to derive the sample complexity of semi-supervised learning with mixture models, when the unsupervised mixture model may not (exactly) match the mixture model implied by the class conditional distributions of the underlying true distribution. This leads to a much more general analysis than earlier work in SSL, both considering misspecification of the mixture and more than 2 classes. They propose several methods to recover the true mapping of decision regions to classes, for which they show both the sample complexity and show empirical results of the probability of correct recovery in three example simulations. # Main comments I would like to thank the authors for this interesting work that further advances our understanding of sample complexities when using mixture model in a semi-supervised setting. While I like the analysis presented in the work, I have some questions, concerns and suggestions whose resolution might help me appreciate the work more. First off, I would suggest reflecting the focus on mixture models in the title, to make it clearer what to expect. Secondly, I have some confusion over the quantify of interest in the analysis. The goal is recovering pi^*. This is perfectly clear in the Lambda=Lambda^* setting, as was also considered in earlier work. However, it is never made precise in the paper (or I missed it) what pi^* represents if Lambda!=Lambda^*. Is it simply the best possible mapping for that particular Lambda (so like (4) but for the expected log likelihood)? Aside from this, how crucial is the restriction that the mixture contains the same number of components as the number of classes? Would it not be enough for the number of components to be bigger than K, or is this not a problem because the components themselves can be arbitrarily complex? Third, I wonder if you could comment on the following. It seems that up except for section 4.4, the analysis does not really depend on being in a semi-supervised setting: regardless of where the mixture model comes from, your results that you can find a mapping from components to classes efficiently continues to hold. In this section, however, the rate in terms of the number of unlabeled samples is rather vague (but still more insightful than papers that only consider clairvoyance). Is there more that can be said about this rate. And for reasonable discrepancies between Lambda and Lambda^* (what are reasonable values for W_1), is it reasonable to assume that the bound in Theorem 4.7 becomes informative? Fourth, I have three questions regarding the experiments. 1. What do the curves look like in terms of classification performance (perhaps these could be added to the appendix, including a supervised baseline). 2. Some of the curves seem to flatten out even though the bounds would suggest that they correct mapping should be obtained with high probability with some number n of samples. Could you explain the relationship between the results in the bounds and the curves (is n simply very large in these cases?). Finally, the paper has a rather abrupt ending. I understand there are space constraints, but it would be nice if the paper could end with something like a concluding sentence. In summary, I find the pros of the paper that it: offers an interesting extension to earlier analyses of mixture models in ssl; is clearly written; has a proper analysis and explanation of the differences between the proposed approaches. I find the cons that: as in earlier work, the focus is mostly on the labeled part of ssl, while the unlabeled part (section 4.4) is less extensive. I am looking forward to the author's response to my questions to better inform my opinion of the paper. # Minor suggestions 15: MLE is not yet defined here 18: "more and more expensive": I'm not sure labeling data has gotten more expensive, labeling all the data might have gotten more expensive because there is more of it. 85: from this definition of M_K it is unclear to me why its elements also contain the lambda_k's, instead of just the f_k's. 101: supervised -> semi-supervised would also make sense here 138: "which is a more difficult problem": in what way? 194: the simplest 231: not -> note Figure 2: Perhaps the legibility of the figure can be improved by making the figure slightly bigger, for instance by sharing the axis annotations of the small multiples. # After response Thank you for your response. Perhaps I had not made clear enough what my confusion with pi^* was in my review, since your response does not specifically address my confusion (or I simply don't understand it): While I understand pi^* is an interesting mapping to estimate if we have Lambda^*, if we have some \hat{Lambda} from the unlabeled data, what is the mapping we are trying to estimate (not necessarily pi^* I suppose), and how does it relate to pi^*. Hopefully this clarifies my question. I would also like to reiterate my suggestion to include "mixture models" or something related in the title to make the goal of the paper clearer up front. Overall, I still find it an interesting paper but I am not more convinced of its merits than I already was before the response.

Reviewer 2



The authors provide a bound for sample complexity in a non parametric setting for multi-class classification task using semi supervised learning. They introduce assumptions based on the differences between the mixture distribution learned from unlabeled data and the true mixture model induced by class conditional distributions. It is good that they tackle the different sub-problems in a systematic manner. They learn decision boundaries for the classifier using the unlabeled data. The labeled data is used to associate the regions with classes. They utilize MLE and Majority vote based estimators to learn the decision region -> actual class mapping. They also provide bounds for classification errors based on the learnt mixture model (based on literature) and the region->class mappings (permutation) Furthermore, They propose algorithms for utilizing MLE to learn the permutation. I am also curious about the following: Are there any other estimators that the authors think might be of interest in practice. Any thoughts about jointly learning both the mixture model and the permutation. The focus of this work seems to be on theoretical contributions. It would be interesting to know how the proposed mechanisms work in practice with a variety of datasets in different domains where the underlying distributions can be much more complex.

Reviewer 3



The paper presents a theoretical analysis of a particular class of semi-supervised algorithms. In this setting the unlabeled data is used to estimate a mixture model $F$ defined by it's mixing measure $\Lambda$; under the assumption that the number $K$ of mixture components is equal to the number of classes (which is a rather strong assumption), the labeled data data is then used to find a permutation $\pi$ of $\{ 1 \dots K \}$ which minimizes the loss of the classifier $g_{\Lambda,\pi} : \mathcal{X} \rightarrow \mathcal{Y}$. In particular the paper examines both the maximum likelihood and majority vote estimators. The presented work decouples (lines 118-122) the analysis of $\Lambda$ focusing on the second step of the algorithm  (lines 77 -81), namely the estimation of $\pi$ given $\Lambda$ and provide sample complexity bounds that relate $P(\hat{\pi} = \pi^*)$ to the number of classes, and mixture components, $K$ and the 'gap' $\Delta$ (equation (6),(7)).  The authors show how the problem of finding the optimal perturbation for a given $\Lambda$ can be, in the case of MLE, be formulated as a bipartite graph matching problem, and be solved using the Hungarian Algorithm, also presenting a greedy, approximate, algorithm for solving the problem (BP-matching). Finally an empirical analysis of the three algorithms is presented in section 6 using both synthetic data (Gaussian) and 'real' data (MNIST). The paper is generally well written, however I do not feel the authors situated there work well within the prior art. The present work seems to be mainly contrasted with the work of Castelli and Cover, and though some generally grouping of other works is presented in lines 110-122, there is no sense of where exactly the work lays in relations to these  (beyond some general positioning). Is the present work the only work that analyses $\pi$ and its relation $\Lambda$?  Of interest here would also be whether or how known SSL algorithms can be analyzed  within this framework. Partly in view of this I struggle to assess the significance of the presented results. It would seem to me that the authors are able to analyze a more general class of mixture models without underlying assumptions, exactly because they eschew analyzing the estimation of $\Lambda$ (and its relation to $\Lambda^*$) and focus on $\pi$ for a given $\Lambda$. Lines 158-160 would seem to support this argument.  But even given this, I am still unsure of the significance of the elicited conditions on $\Lambda$. The authors mention that the 'ease' of learning $\pi$ is related to the quantity $\Delta$ and mention that a 'crucial condition' (line 152) is $\Delta>0$.  Here I am not sure I understand. It is obvious that for $\Delta \leq 0$, the bound in Theorem 4.1 becomes meaningless, however it is not clear to me that this translates to it being a necessary condition. Perhaps I am not understanding the meaning of 'crucial' but I feel this is an important point, if there is some prior art on how 'learnability' relates to the gap $\Delta$ this would be helpful, particularly given the fact that the condition $\Delta>0$ is subsequently accepted as a 'weak' assumption. It is also not clear why this assumption is considered 'weak'.  Proposition 4.2 does not help back up this claim as it based on the assumption that $\Lambda = \Lambda^*$ which is a very strong, in fact unrealistic, assumption. I have similar concerns regarding the analysis in section 4.2. Regarding section 4.3, could the authors comment on the assumption $\Delta \in \Omega(1)$, the assumption that the gap is not related to $K$ the number of mixture components/classes does not make intuitive sense to me. Is there perhaps prior art here? Finally, as the main focus and strength of the paper is the theoretical analysis, I do not feel the empirical analysis, though interesting, provides any particular insight into the analysis. It could perhaps be better fitted in the appendix with more space in the main paper to analyze the theoretical results. # Post response In response to authors' feedback and their willingness to revise the paper as per reviewers' input, I am updating my overall score for this submission.